# Marine-Derived Natural Substances with Anticholinesterase Activity

**DOI:** 10.3390/md23110439

**Published:** 2025-11-15

**Authors:** Daniela Dimitrova, Gabriela Kehayova, Simeonka Dimitrova, Stela Dragomanova

**Affiliations:** 1Faculty of Pharmacy, Medical University of Varna, Tsar Osvoboditel Blv. 84A, 9000 Varna, Bulgaria; danielladad@icloud.com; 2Department of Pharmacology, Toxicology, and Pharmacotherapy, Medical University of Varna, Tsar Osvoboditel Blv. 84A, 9000 Varna, Bulgaria; gabriela.kehayova@mu-varna.bg (G.K.); simeonka.dimitrova@mu-varna.bg (S.D.)

**Keywords:** neuroprotection, anti-cholinesterase, antioxidant, marine natural products, algae, fungi, bacteria, marine invertebrates, fish

## Abstract

Alzheimer’s disease continues to be one of the most urgent neurodegenerative conditions, with acetylcholinesterase (AChE) inhibitors serving as a fundamental component of contemporary treatment approaches. Growing evidence underscores that marine ecosystems are a rich source of structurally varied and biologically active natural products exhibiting anticholinesterase properties. This review presents a thorough synthesis of marine-derived metabolites—including those sourced from bacteria, fungi, sponges, algae, and other marine life—that demonstrate inhibitory effects against AChE and butyrylcholinesterase (BuChE). Numerous compounds, such as meroterpenoids, alkaloids, peptides, and phlorotannins, not only show nanomolar to micromolar inhibitory activity but also reveal additional neuroprotective characteristics, including antioxidant effects, anti-amyloid properties, and modulation of neuronal survival pathways. Despite these encouraging findings, the transition to clinical applications is hindered by a lack of comprehensive pharmacokinetic, toxicity, and long-term efficacy studies. The structural variety of marine metabolites provides valuable frameworks for the development of next-generation cholinesterase inhibitors. Further interdisciplinary research is essential to enhance their therapeutic potential and facilitate their incorporation into strategies for addressing Alzheimer’s disease and related conditions.

## 1. Introduction

Alzheimer’s disease (AD) is a progressive neurodegenerative disorder characterized by significant memory impairment, disrupted synaptic activity, and neuronal loss. Despite extensive research spanning several decades, a singular universal cause of the disease remains unidentified; rather, a multitude of hypotheses exist that seek to elucidate the underlying pathogenesis. In recent years, there has been an increasing necessity to understand the complex mechanisms that initiate the pathological processes linked to the disease. The most frequently debated hypotheses include those related to cholinergic dysfunction, proteinopathies, neuroinflammation, as well as vascular, metabolic, infectious factors, among others (Figure 1).

The early degeneration of the cholinergic (acetylcholinergic) neuronal system—particularly from the basofrontal regions that supply ACh to the cortex and hippocampus—plays a crucial role in cognitive deterioration. The cholinergic system is integral to memory functions, with acetylcholine serving as a vital mediator for information processing and the establishment of memory traces. A reduction in cholinergic neurotransmission can result in memory deficits and disorientation, which are commonly observed in AD and other neurodegenerative disorders [1].

AD is marked by the degeneration of cholinergic neurons, a reduction in the neurotransmitter ACh, and the activation of the enzyme that degrades it in the synaptic cleft, known as AChE. This neurodegenerative process is linked to cognitive impairments and a notable decline in cholinergic neurons, typically found in brain regions that govern memory and learning, including the hippocampus, cortex, and the basal nucleus of Meyner [2,3]. The reduction in cholinergic innervation is associated with the extent of cognitive decline) [4].

The cholinergic mechanism, particularly the heightened activity of AChE and the reduction in cholinergic innervation, should not be viewed as a solitary pathogenetic hypothesis. Instead, it is interconnected with other processes, including inflammation, vascular injury, and metabolic disorders, and interacts with other prominent hypotheses regarding the pathogenesis of AD.

In line with the amyloid hypothesis, the buildup and aggregation of the Aβ peptide within the brain initiates a series of pathological events, including the development of amyloid plaques, the activation of inflammatory responses, and neurodegeneration. This is supported by clinical evidence derived from treatments involving novel drugs, which are antibodies targeting amyloid plaques [5]. Nevertheless, genetic, clinical, imaging, and biochemical evidence indicates a more intricate etiology of neurodegenerative processes. This is corroborated by the capacity of activated AChE to facilitate the development of amyloid plaques by hastening the aggregation of amyloid-beta peptides into harmful fibrils, as AChE-Aβ complexes exhibit greater toxicity than Aβ fibrils in isolation [6]. AChE is colocalized within Aβ deposits and promotes the expression of Aβ-precursor protein while also activating glial cells, leading to an increase in neuroinflammation that further aids in the development of plaques [7]. The dual function of AChE plays a significant role in both cholinergic dysfunction and the accumulation of amyloid plaques [8].

The tau hypothesis posits that tau proteins, when pathologically phosphorylated, aggregate to form neurofibrillary tangles that interfere with neuronal transport and structural integrity, which is closely associated with cognitive decline in AD. This hypothesis is frequently regarded as complementary to the amyloid cascade, as amyloid deposits are believed to trigger tau hyperphosphorylation, consequently leading to neuronal damage [9]. The resultant tau tangles exacerbate the detrimental effects of beta-amyloid. This interaction establishes a positive feedback loop whereby amyloid promotes tau hyperphosphorylation, resulting in an accelerated progression of neurodegeneration. The simultaneous presence of amyloid plaques and tau tangles significantly disrupts neuronal communication and functionality [10]. This results in the loss of synapses, the death of neurons, and considerable cognitive decline. Furthermore, heightened activity of cholinergic enzymes, such as AChE, is associated with the buildup of neurofibrillary tangles and inflammatory markers during the initial phases of dementia [11].

Neuroinflammation refers to the persistent inflammation occurring within the central nervous system, which facilitates neurodegeneration. Activated microglia, the immune cells of the brain, are pivotal in this process. These microglia are activated by signals originating from injured neurons or various other stimuli, leading them to secrete pro-inflammatory substances (including IL-1β and TNF-α), produce oxidative stress, and potentially assault neurons. This results in a sustained, pro-inflammatory environment that harms adjacent neurons [12,13]. The cholinergic system plays a crucial role in regulating inflammatory signals through what is referred to as the “cholinergic anti-inflammatory pathway”—for instance, via α7-nicotinic acetylcholine receptors [14]. In AD, disrupted cholinergic signaling leads to the activation of microglia and astrocytes, as well as an elevated expression of inflammatory cytokines, thereby connecting cholinergic deficiency to the neuroinflammatory hypothesis [15].

The hypothesis of metabolic dysfunction, often referred to as “type 3 diabetes,” posits that insulin resistance, compromised glucose metabolism, and mitochondrial dysfunction could significantly impact the onset of AD by means of disrupted energy provision, inflammatory processes, and the buildup of abnormal proteins [16,17]. In the realm of metabolic changes (such as insulin resistance and type 2 diabetes), a deficiency in cholinergic function is believed to intensify metabolic stress (for instance, via compromised neuronal metabolism, reduced energy levels, and heightened oxidative stress). This underscores the connection between the cholinergic hypothesis and the metabolic hypothesis of AD. Insulin resistance interferes with brain metabolism, promotes inflammatory processes, and engages with Aβ/tau pathways [18,19]. The consequent ACh-mediated synaptic dysfunction and energy deficiency connect the metabolic hypothesis to cholinergic disorders. Mitochondrial dysfunction, compromised mitophagy, and heightened oxidative stress result in energy deficits and neuronal cell death, which is incorporated into the metabolic hypothesis. This process is frequently regarded as a “bridge” linking metabolic disorders, vascular injury, neuroinflammation, and traditional Aβ/tau pathology.

An alternative, yet progressively popular hypothesis is the infectious one. This theory posits that different infections (bacterial, viral, fungal) may initiate or hasten pathological alterations, including neuroinflammation or the buildup of Aβ (which could act as an antimicrobial peptide). The review conducted by Vojtechova et al. (2022) examines the influence of microorganisms and portrays Aβ as a component of the antimicrobial defense system [20], while Onisiforou et al. (2025) examine the significance of intestinal flora, the oral microbiome, viruses, and brain biofilms [21].

The vascular hypothesis posits that conditions affecting the vascular system, such as chronic cerebral ischemia, hypertension, diabetes, and various other vascular risk factors, elevate the likelihood of developing AD [22,23]. The analysis conducted by Eisenmenger et al. in 2023 highlights that vascular risk factors and vascular pathology frequently coexist with AD and may either initiate or exacerbate the condition [24]. Regarding vascular effects, disrupted cholinergic signaling may influence cerebral blood flow, vasomotor regulation, and the integrity of the blood–brain barrier. The cholinergic pathway serves as a link between vascular risks and neuroinflammatory processes [25]. A meta-analysis examining the application of cholinesterase inhibitors in cases of vascular cognitive impairment reinforces the clinical association between cholinergic treatment and the vascular aspects of cognitive deficits [26].

Current research substantiates the perspective that AD is a multifactorial condition, characterized by various interacting pathogenetic pathways, including neuromodulatory, vascular, metabolic, inflammatory, and infectious factors, rather than being attributed to a singular causative agent. Consequently, it is probable that the various hypotheses elucidate components of the overarching process that culminates in neurodegeneration. Understanding these hypotheses and their relationships is essential for further developing effective strategies for prevention and treatment. [27].

Conversely, while Alzheimer’s disease is examined through various pathogenetic theories, cholinergic deficiency, especially the function of acetylcholinesterase, emerges as a crucial linking mechanism among them (as illustrated in Figure 1). Regulation of this enzyme’s activity not only influences cognitive symptoms but also is intricately associated with inflammatory processes, amyloidogenesis, and neuronal metabolism [28]. This underscores the necessity of exploring novel acetylcholinesterase inhibitors that integrate anticholinergic properties with supplementary advantageous effects aimed at other pathogenic mechanisms, thereby paving the way for more effective and multimodal therapeutic approaches.

## 2. Results

### 2.1. Marine Bacteria as a Source of Anticholinesterase Compounds

Marine bacteria, especially those linked to sponges and corals, have surfaced as a potential source of anticholinesterase compounds. These microorganisms generate a range of bioactive metabolites, such as pyrrole derivatives and various other secondary metabolites. The examination of marine bacteria, especially those linked to marine invertebrates, has shown that a considerable proportion of isolates demonstrate AChE inhibitory activity. For example, a strain of *Bacillus subtilis* obtained from the sponge *Fasciospongia cavernosa* displayed significant enzyme inhibition [29]. The extract derived from this strain demonstrated considerable AChE inhibitory activity, and two major components (Rf = 0.45 and Rf = 0.85) were detected by TLC bioautography as potential novel inhibitors, distinct from galantamine, although their chemical structures remain to be identified.

Additionally, *Streptomyces lateritius* has been recognized as a source of pyrrole-derived compounds that possess significant AChE inhibitory properties [30]. Research indicates that these compounds can effectively scavenge free radicals and demonstrate minimal cytotoxic effects, rendering them promising candidates for pharmaceutical development.

### 2.2. Marine Fungi as a Source of Anticholinesterase Compounds

Marine fungi have shown significant potential as sources of anticholinesterase compounds, which are crucial for treating neurodegenerative diseases like AD by inhibiting AChE and enhancing cholinergic transmission. Marine fungi, particularly those associated with corals and sponges, have been identified as rich sources of AChEIs. For instance, the fungus *Aspergillus ochraceus* produced circumdatin D, which demonstrated potent AChE inhibition and neuroprotective effects in AD-like nematode models by delaying paralysis and reducing inflammation. The compounds from marine fungi often exhibit multi-target strategies, not only inhibiting AChE but also modulating inflammatory pathways. Circumdatin D, for example, interferes with Toll-like receptor 4 (TLR4)-mediated NF-κB, MAPKs, and JAK/STAT pathways, providing a comprehensive neuroprotective effect [31]. *Daldinia eschscholtzii* isolated from stony corals, showed significant AChE inhibition, with its metabolites 4,7-dihydroxycoumarin and 5-nitro-2-naphthalenamine identified as active compounds. Through GC-MS analysis and molecular docking, they identified key compounds likely responsible for AChE inhibition and proposed binding interactions with the enzyme’s active site. Their findings present *D. eschscholtzii* as a promising source of cholinergic modulators for neurodegenerative disease research [32]. A 2020 study explored marine-derived *Aspergillus terreus* and revealed its capacity to produce diverse AChEIs, notably territrem analogues. Using a combination of bioassays, LC–MS/MS profiling, and Global Natural Products Social Molecular Networking, they successfully mapped the chemical diversity of its metabolites. These compounds displayed both AChE inhibitory and antioxidant activities, highlighting their dual role in enhancing cholinergic transmission and reducing oxidative stress, two central mechanisms in neurodegenerative disease progression. The study underscored *A. terreus* as a valuable marine source of multifunctional metabolites with therapeutic potential [33]. Preclinical research on the metabolites of marine fungi has shown great promise in neurodegeneration models. Strong AChE inhibitory activity and notable antioxidant effects were observed in extracts and compounds that were isolated from *Aspergillus sydowii* and *Penicillium chrysogenum* [34,35]. Both neurotoxicity and oxidative stress, two key mechanisms behind the development of neurodegenerative diseases like Alzheimer’s, were quantifiably reduced because of these two actions. Crucially, these metabolites’ combined AChE inhibitory and antioxidant qualities increase their therapeutic value by enhancing cholinergic neurotransmission and shielding neurons from damage caused by reactive oxygen species. The idea that marine-derived fungal compounds act as multitarget agents with potent neuroprotective potential is supported by previous screenings of marine fungi, which similarly identified anti-Alzheimer’s bioactive constituents with these overlapping mechanisms [36].

The identification of compounds exhibiting anticholinesterase activity from diverse natural sources has been a subject of research interest for an extended period. Three meroterpenes—arisugacins B-D—were extracted from the endophytic fungus *Penicillium* sp. FO-4259 [36,37]. In vitro studies revealed that these compounds displayed remarkable inhibitory effects on AChE, with IC_50_ values of 25.8 nM, 2.5 µM, and 3.5 µM, respectively. Furthermore, arisugacin A, a highly selective AChE inhibitor derived from *Penicillium* sp. FO-4259, demonstrated a significant binding affinity with an IC_50_ of 1.0 nM. The anti-cholinesterase activities of arisugacins D, M, O, P, and Q were confirmed in a different study [38]. Among these, arisugacin O exhibited the highest inhibitory potency in vitro, with an IC_50_ value of 191 nM. However, its application to zebrafish embryos resulted in paralysis.

Arisugacins B, E, F, and I, along with dehydroaustinol and isoaustinone, were recognized as promising candidates for anti-cholinesterase activity in vitro [39]. Additionally, 2017 study demonstrated that the meroterpenoids isoaustinol, dehydroaustin, and dehydroaustinol inhibit AChE, with IC_50_ values of 2.50, 0.40, and 3.0 µM, respectively [40]. The subsequent chemical analysis of the mangrove-derived fungus *Penicillium* sp., sourced from the foliage of the mangrove species *Kandelia candel*, resulted in the identification of several novel compounds: 3-epi-arigsugacin E, terreulactone C, territrem B, territrem D, and territrem E [41]. This finding complements the previously identified arisugacin B and arisugacin D. The newly discovered compounds (arisugacin B, territrem C, and terreulactone C) exhibited significant inhibitory activity against AChE, which is particularly noteworthy. It has also been established that terreulactone A [42] and isoterreulactone A [43] possess anti-cholinesterase properties. The effects of both substances are dose-dependent. Amphichoterpenoids D and E, two meroterpenoids isolated from *Amphichorda felina* SYSU-MS7908, demonstrate in vitro anti-cholinesterase activity with IC_50_ values of 12.5 μM and 11.6 μM, respectively [44]. The marine-derived fungus *Aspergillus versicolor* also showed moderate inhibitory activity against AChE, with an IC_50_ value of 13.6 μM [45]. In 2019, Luo et al. discovered five novel meroterpenoids derived from *Ganoderma lucidum* [46]. Notably, dayaolingzhiols D and E displayed significant AChE inhibitory effects, with IC_50_ values of 8.52 ± 1.90 μM and 7.37 ± 0.52 μM, respectively. Isolated from the marine-derived fungus *A. terreus*, territrem B exhibited significant anticholinesterase activity with an IC_50_ value of 0.071 μM, indicating it as a potent inhibitor [47].

Our earlier research summarized the neuroprotective capabilities of fungal meroterpenoids, including their AChE inhibitory activity [48]. A variety of metabolites isolated from *Penicillium* and *Aspergillus* species demonstrated significant inhibition, often within the nanomolar to low micromolar range. Among the most effective were territrem B and territrem C from Aspergillus terreus, along with arisugacin A (IC_50_ ≈ 1 nM) and arisugacin B (IC_50_ ≈ 3 nM) from *Penicillium* spp. Additionally, terreulactone A, cyclophostin, and dehydroaustinol were identified as having considerable inhibitory effects, although their impact was generally less significant. The structural integration of polyketide and terpenoid components in these meroterpenoids facilitates various interactions within the AChE catalytic site. A clinically significant aspect that may reduce cholinergic side effects is the selectivity for AChE over BuChE, which was also evidenced by some compounds. Beyond enzyme inhibition, several compounds displayed antioxidant and anti-amyloid activities, suggesting a multi-target neuroprotective mechanism. However, most findings are based on in vitro studies, highlighting the need for further research into in vivo efficacy, toxicity, and pharmacokinetics before clinical application.

In addition, the fungus *A. niger* has been shown to produce naphthoγ-pyrones and an alkaloid, namely aurasperone A, fonsecinone A, and aspernigrin A [49]. The naphtho-γ-pyrone flavasperone and aurasperone A are known to inhibit AChE activity. Notably, the environmentally friendly synthesis of silver nanoparticles (AgNPs) utilizing these compounds markedly improved their inhibitory activity, with certain compounds exhibiting potency that is comparable to or exceeds that of the reference drug galanthamine (IC_50_ value of 1.43 μM). For example, the inhibitory effect of fonsecinone A was found to increase by 84-fold following the synthesis of AgNPs, with its IC_50_ value decreasing from 7.52 μM to 0.089 μM. In a similar manner, the activity of aurasperone A rose by 16 times, while the activity of aspernigrin A increased by 13 times. The study also emphasized that the mechanism through which AgNPs inhibit AChE may involve structural alterations in the enzyme resulting from the adsorption of nanoparticles onto its surface. This interaction could potentially amplify its inhibitory effects. However, the paper acknowledges certain limitations. While the potential neurotoxicity of AgNPs is mentioned, the research lacks comprehensive long-term toxicity studies or evaluations regarding the safety of these nanoparticles within biological systems. Moreover, the authors suggest that the combination of AgNPs with naphtho-γ-pyrones enhances AChE inhibition; nevertheless, they did not perform an exhaustive investigation into the mechanisms that underlie these synergistic effects. In summary, the paper offers significant insights into the structure-activity relationships of the isolated compounds, suggesting that specific structural characteristics can enhance AChE inhibition. Furthermore, the marine fungus *P. chrysogenum* has been reported to produce metabolites exhibiting AChE inhibitory activity [35]. The extract derived from this fungus demonstrated an inhibition percentage of 63% and an IC_50_ value of 60.87 µg/mL, underscoring its potential as an AChE inhibitor of natural origin.

Marine fungi are promising sources of anticholinesterase compounds with significant ability to inhibit AChE and modulate inflammatory pathways. This positions them as potential candidates for developing new treatments for neurodegenerative diseases like AD.

### 2.3. Marine Sponges as a Source of Anticholinesterase Compounds

Marine sponges generate a diverse array of secondary metabolites, such as alkaloids and terpenoids, which have been investigated for their enzyme-inhibitory characteristics. Marine sponges have proven to be a valuable source of bioactive metabolites exhibiting anticholinesterase activity.

Barettin and its derivatives, extracted from the sponge *Geodia barretti*, have demonstrated the ability to inhibit AChE via a reversible noncompetitive mechanism [50]. These compounds exhibit inhibitory constants that are comparable to those of other natural AChEIs, underscoring their potential as therapeutic agents. Additionally, synthesized brominated indole derivatives have been evaluated for their anticholinesterase activity. Although these compounds display moderate inhibitory effects, their structure-activity relationships offer significant insights for the development of synthetic analogs with enhanced potency.

Another significant instance is iso-agelasine C, a diterpene alkaloid derived from the marine sponge *Agelas nakamurai*. This compound demonstrated moderate inhibitory activity against AChE, positioning it as a potential candidate for subsequent drug development [51]. Iso-agelasine C presented an IC_50_ value of 30.68 ± 0.92 µg/mL, reflecting its efficacy as an AChE inhibitor. Nevertheless, this value is inferior when compared to the extracts and fractions obtained from the sponge, implying that the synergistic effects of various compounds within the extracts may yield stronger inhibition. Additionally, the results indicate that iso-agelasine C could act as a lead compound for the further advancement of halimane alkaloids as AChE inhibitors. The study notes that the enhanced AChE inhibitory activity observed in the extracts and fractions may stem from the synergistic interactions of multiple compounds. However, the paper lacks a comprehensive analysis or data regarding the interactions among these compounds, which could be crucial for comprehending the overall effectiveness of the sponge extracts. While the research points that structural alterations of iso-agelasine C might improve its AChE inhibitory activity, it does not investigate or suggest specific modifications. This omission creates a gap in the research concerning the optimization of the compound for improved therapeutic results.

Recently, the alkaloid discorhabdin G, extracted from the Antarctic sponge *Latrunculia*, has been investigated for its potential role as a cholinesterase inhibitor. Utilizing molecular docking and enzymatic assays, it was determined that discorhabdin G functions as a reversible competitive inhibitor of AChE, exhibiting greater selectivity for AChE in comparison to BuChE [52]. The study employed molecular docking to propose the pharmacophore moiety of discorhabdin G. This methodology enabled the researchers to streamline the structure of the metabolite, resulting in the identification of a more promising candidate molecule, 5-methyl-2H-benzo[h]imidazo [1,5,4-de]quinoxalin-7(3H)-one. The candidate molecule was assessed for its inhibitory effects on various acetylcholinesterases, including electric eel AChE (eeAChE), human recombinant AChE (hAChE), and horse serum BuChE. It displayed a slightly reduced inhibitory capacity against eeAChE but showed enhanced activity against hAChE when compared to discorhabdin G. The new compound demonstrated increased selectivity for AChEs over BuChE, functioning as a reversible competitive inhibitor, akin to the natural alkaloid.

### 2.4. Marine Algae as a Source of Anticholinesterase Compounds

Marine algae have been recognized as a valuable source of bioactive compounds that hold considerable promise for neuroprotection, especially due to their anticholinesterase activity. These compounds act by inhibiting AChE and BuChE that may contribute to the management of neurodegenerative diseases.

Compounds such as sargaquinoic acid and sargachromenol derived from *Sargassum sagamianum* exhibited moderate inhibitory activity against AChE, with sargaquinoic acid demonstrating notable potency against BuChE [53]. Sargaquinoic acid, which is derived from the brown alga *S. sagamianum*, exhibited moderate inhibitory activity against AChE (IC_50_ 23.2 μM) but was particularly effective against BuChE, with an IC_50_ of 26 nM, positioning it as a promising candidate for the treatment of AD. Another compound, 6,6′-bieckol, extracted from the red alga *Grateloupia elliptica*, also displayed moderate AChE inhibitory activity (IC_50_ 44.5 μM) alongside potent BuChE inhibitory activity (IC_50_ 27.4 μM) [54].

Phlorotannins derived from the edible algae *Ecklonia stolonifera*, found in the Sea of Japan, such as phlorofucofuroeckol-A, dieckol, and eckol demonstrated significant inhibitory effects against AChE, with IC_50_ values of 4.89, 17.11, and 20.56 μM, respectively [55]. Fucosterol, 24-hydroperoxy 24-vinylcholesterol, eckstolonol, and phlorofucofuroeckol-A also showed activity against BuChE.

Additionally, ethanolic extracts from the seaweed *E. bicyclis* revealed promising inhibitory properties against both AChE and BuChE, with the ethyl acetate fraction demonstrating the most potent activity [56].

A 2015 research article examined the potential of organic extracts from three Brazilian red algae—*Hypnea musciformis*, *Pterocladiella capillacea*, and *Ochtodes secundiramea*—as inhibitors of AChE [57]. The study utilized Ellman’s microplate assay, revealing that the crude extract of *O. secundiramea* exhibited moderate inhibitory activity, achieving around 48% inhibition at a concentration of 400 µg/mL, whereas the other two species displayed only minimal activity. The analysis using gas chromatography-mass spectrometry (GC-MS) showed that the bioactive fraction of *O. secundiramea* consisted exclusively of halogenated monoterpenes, including derivatives of myrcene, pinene, and various other typical monoterpene structures. Authors assumed that these brominated and chlorinated terpenoids interact with the active site of AChE through hydrophobic and halogen-bond interactions, thereby obstructing the enzymatic breakdown of ACh. Toxicological assessments showed no cytotoxic or mutagenic effects at the concentrations tested, underscoring their favorable safety profile for prospective pharmacological research.

In a different study from 2013, researchers explored the AChE inhibitory effects of phlorotannins extracted from the brown kelp *Ecklonia maxima* [58]. This represents the inaugural report of cholinesterase inhibition associated with this species. Employing a modified Ellman assay, the researchers analyzed crude methanolic extracts, solvent fractions, and purified phlorotannins, including derivatives of phloroglucinol such as eckol. The findings indicated moderate inhibition, with IC_50_ values ranging from 62.6 µg/mL to 150.8 µg/mL, contingent upon the fraction or isolated compound evaluated. Phlorotannins, which are marine-specific polyphenols composed of phloroglucinol units interconnected by aryl-aryl and aryl-ether bonds, are recognized for their potent antioxidant properties. These properties may synergistically enhance AChE inhibition, thereby offering dual neuroprotective benefits.

Marine macroalgae have yielded a wide array of cholinesterase inhibitors. For example, the green alga *Capsosiphon fulvescens* [59] produces glycolipid metabolites (capsofulvesins A–B) that inhibit electric eel AChE with IC_50_ ≈ 50–53 μM (diacyl forms) versus ~ 80 μM for monoacyl analogues. Brown algae (e.g., *Ecklonia* spp.) furnish phlorotannin polyphenols: phlorofucofuroeckol A inhibits AChE (IC_50_ ≈ 4.89 μM), and related polyphenols (e.g., 8,8′-bieckol) inhibit both AChE and BChE (IC_50_ ~ 16.0 and 10.9 μM); these bulky molecules appear to sterically block the enzyme gorge (non-competitive inhibition). Red algae similarly provide inhibitors: *Gelidiella acerosa* [60] contains phytol (a diterpene) that inhibits AChE (IC_50_ = 2.7 μM) and BChE (5.8 μM), and halogenated bromophenols from *Rhodomela* and *Polysiphonia* spp. exhibit IC_50_ values in the low picomolar range. Inhibitory activity, typically assayed by the Ellman method using eeAChE or equine BChE, and mechanistic studies (e.g., docking of phytol) suggest binding to peripheral residues such as Arg177 outside the catalytic site. These examples illustrate that algal metabolites (glycolipids, polyphenols, terpenes, halophenols) can modulate cholinesterase activity, with structural features (chain length, polymerization, halogenation) influencing potency. Extracts derived from different species of algae, such as *Gracilaria corticata* [61], have shown significant AChE inhibitory activity in preclinical research [62]. Although many preclinical investigations have shown the neuroprotective properties and cholinesterase inhibitory effects of marine algae, there remains a scarcity of clinical trials. Nevertheless, the authorization of sodium oligomannate, an oligosaccharide derived from algae, for the treatment of AD represents a noteworthy advancement [63]. Initial clinical studies indicate that the intake of seaweed could potentially aid in the prevention of cerebrovascular diseases and enhance cognitive abilities by modulating neurotransmitter levels and diminishing neuroinflammation.

The red alga *Gracilaria manilaensis* has been identified to possess neuroprotective compounds, including cynerine A and graveolinine, which demonstrate inhibitory activities against both AChE and BuChE. Research indicates that these compounds function through mixed and uncompetitive inhibition mechanisms, thereby emphasizing the therapeutic potential of substances derived from algae [64]. Both the methanol extract (methanolic extract, prepared by maceration, GMM) and its most potent fraction (fraction G obtained from GMM through column chromatography, GMMG) significantly inhibited AChE, with GMMG demonstrating the highest effectiveness (EC_50_ = 2.3 mg/mL). Enzyme kinetic studies indicated that GMM functioned through a mixed inhibition mechanism, whereas GMMG displayed uncompetitive inhibition by binding solely to the enzyme–substrate complex. This particular modification of AChE, with minimal impact on BuChE, could be beneficial for AD treatment, as excessive inhibition of BuChE has been linked to cholinergic side effects. The robustness of the study is attributed to its mechanistic enzyme kinetics analysis; however, the lack of in vivo validation constrains the ability to draw conclusions regarding therapeutic efficacy and safety.

Recently, compounds like diphlorethohydroxycarmalol and phlorofucofuroeckol derived from brown seaweeds have indicated potential as AChEIs. These compounds were discovered through molecular docking studies and were observed to interact with crucial residues in the active site of the enzyme, positioning them as promising candidates for AD therapy [65]. The study concentrated on the inhibition of two crucial enzymes, AChE and BuChE, which play a role in the hydrolysis of acetylcholine. It identified various bioactive compounds derived from brown, red, and green algae, recognized for their neuroprotective properties and potential health advantages, including antioxidant effects and significant nutritional value. The researchers performed molecular docking studies, molecular dynamics simulations, and binding energy assessments to analyze the interaction between the identified compounds and the target enzyme AChE. The findings underscored three bioactive compounds that exhibited the highest binding affinity to AChE. Among these compounds, diphlorethohydroxycarmalol and phlorofucofuroeckol have been identified as the most promising candidates for further advancement in the treatment of neurodegenerative disorders. While the study offers encouraging results regarding compounds derived from marine algae for the treatment of neurodegenerative diseases, several limitations should be acknowledged. The research predominantly relied on in silico methods; although these techniques are useful for predicting interactions, they fail to consider the intricate biological environment present in living organisms. Experimental validation within biological systems is essential to confirm the efficacy of the identified compounds. Moreover, the research concentrated on a limited number of bioactive compounds from brown, red, and green algae. This narrow focus may overlook other potentially beneficial compounds found in marine algae that could also demonstrate neuroprotective effects. A more extensive screening of various algae species and their compounds could uncover additional candidates.

### 2.5. Compounds with Anticholinesterase Activity Derived from Other Marine Organisms

Besides bacteria, sponges, and algae various marine organisms including tunicates, corals, crustaceans and fish have also been investigated for their anticholinesterase properties.

Marine invertebrates such as tunicates and corals, as well as other marine organisms, have been recognized as prolific sources of structurally unique metabolites with AChE inhibitory properties. Several classes of compounds have been reported to exert neuroprotective effects relevant to AD and related disorders. For instance, the clinically used alkaloid galantamine, originally derived from natural sources, remains a prototypical example of an AChE inhibitor. Beyond alkaloids, a wide range of marine-derived compounds displays notable activities [66]. Chitosan oligosaccharides (COS) and their derivatives, such as diethylaminoethyl chitosan oligosaccharides (DEAE-COS), have been shown to act as competitive inhibitors of AChE. Sulfated polysaccharides like fucoidans reduce Aβ toxicity and apoptosis in neuronal models, highlighting their multifunctional neuroprotective potential. Polyphenolic phlorotannins, including dieckol, eckol, and 8,8′-bieckol, inhibit both AChE and β-secretase (BACE-1), thereby potentially attenuating Aβ generation, while dieckol also conferred protection against Aβ25–35-induced cytotoxicity in PC12 cells. Marine sterols such as fucosterol demonstrated moderate AChE inhibition in Ellman’s assay and have been identified as noncompetitive inhibitors of human AChE with higher affinity than reference compounds. Carotenoids, including fucoxanthin and astaxanthin, have also been reported to suppress AChE activity, mitigate oxidative stress, and improve cognitive performance in animal models. Collectively, these findings emphasize the structural diversity and therapeutic promise of marine natural products as leads for the development of novel neuroprotective agents targeting cholinesterase activity and associated pathogenic mechanisms in neurodegeneration.

In a separate comprehensive review, Ghoran and Kijjoa systematically catalog marine-derived compounds targeting AD by inhibiting enzymes such as AChE, BuChE, BACE-1, and various kinases, and by modulating Aβ aggregation pathways [67]. Among the alkaloids, marinoquinoline A, isolated from *Rapidithrix thailandica*, showed AChE inhibition with an IC_50_ of 4.9 µM, while other fungal N-methoxyindolediketopiperazine derivatives (acrozines) gave IC_50_ values in the range 9.5–130.5 µM, with stereoisomers varying in potency. Pulmonarins A and B, from the ascidian *Synoicum pulmonaria*, acted as noncompetitive inhibitors of electric eel AChE (IC_50_ = 105 and 36 µM, respectively). The alkaloid fascaplysin, from marine sponges, inhibited both AChE and BuChE (IC_50_ = 1.49 µM for eeAChE, and 90.47 µM for eqBuChE), in a noncompetitive mode. Notably, petrosamine, from a marine sponge extract, exhibited an exceptional IC_50_ of 0.091 µM against eeAChE—approximately sixfold more potent than galantamine. Among phenolics, p-terphenyl derivatives (e.g., 6′-O-desmethylterphenyllin analogues) showed AChE inhibition in the low micromolar range (IC_50_ ≈ 5–8 µM). Terpenoids such as 14-acetoxycrassine and asperdiol, isolated from corals and sea whips, inhibited AChE dose-dependently with IC_50_ values of 1.40 and 0.358 µM, respectively, while phytol (from a red alga) also showed dual AChE/BuChE inhibition. In the sterol class, compounds like cholesterol derivatives and fucosterol (from *Sargassum horridum*) exhibited moderate but notable inhibition, with fucosterol acting as a noncompetitive inhibitor of human AChE. Overall, this review underscores that marine natural products span diverse structural classes and exhibit wide ranges of inhibitory potency and mechanisms, highlighting many promising leads for further development in AD therapeutics [68].

Androtoxin B, a neurosteroidal alkaloid derived from the venom of the upside-down jellyfish *Cassiopea andromeda*, exhibits potent inhibitory activity against AChE [69]. Structural analysis utilizing GC–MS and NMR techniques validated its classification as a C_27_ steroidal alkaloid (MW 405 Da). Concurrent bioassays exhibited substantial inhibitory activity, with an IC_50_ value of 2.24 ± 0.1 µM, surpassing the efficacy of galantamine (IC_50_ = 6.1 ± 0.13 µM). Furthermore, docking studies corroborated these results, indicating a binding energy of −12.31 kcal/mol and significant interactions with essential catalytic residues (Ser203 and His447), in addition to peripheral aromatic residues such as Tyr124, Trp286, and Tyr341. The dual engagement with both catalytic and peripheral sites, along with extensive hydrophobic and hydrogen bonding interactions, emphasizes the robustness of its inhibitory profile.

Crustaceans, including marine crabs and shrimps, have been investigated for their ability to generate neuroprotective compounds. Although they are not as extensively studied as other marine sources, these organisms present a promising direction for future research. Peptides obtained from marine life, including the hexapeptide QMDDQ sourced from shrimp, have demonstrated neuroprotective properties by inhibiting AChE activity and stimulating the protein kinase A/cAMP response element-binding protein/ brain-derived neurotrophic factor (PKA/CREB/BDNF) signaling [70]. This particular peptide has been shown to enhance memory in mice with scopolamine-induced amnesia, thereby reinforcing its potential for therapeutic applications.

In addition to the peptide-based discoveries, investigations into marine fish oils sourced from the northeastern coast of Brazil have uncovered that specific species yield lipid fractions exhibiting significant AChE inhibitory activity. Oils extracted from fish species demonstrated notable AChE inhibitory effects, attributed to their high levels of polyunsaturated fatty acids [71]. Notably, oils derived from *Scomberomorus cavalla* (IC_50_ = 2.60 ± 0.150 μg/mL), *Lutjanus synagris* (IC_50_ = 2.84 ± 0.044 μg/mL), and *Haemulon plumieri* (IC_50_ = 4.81 ± 0.029 μg/mL) demonstrated remarkable efficacy, nearing the activity of the standard inhibitor physostigmine (IC_50_ = 1.15 ± 0.047 μg/mL). Analysis via gas chromatography–mass spectrometry revealed that these oils, while typical of tropical fish due to their high saturated fatty acid content (particularly palmitic acid, 26–45%), also possessed significant levels of monounsaturated fatty acids (MUFAs) (oleic acid, up to 34.95%) and, importantly, considerable quantities of polyunsaturated fatty acids (PUFAs), including eicosapentaenoic acid (EPA) and docosahexaenoic acid (DHA). The elevated PUFA levels were associated with the most pronounced AChE inhibition, reinforcing the importance of omega-3 fatty acids in preserving neuronal membrane integrity, regulating neuroinflammation, and alleviating oxidative stress—all of which are processes pertinent to AD pathology.

Table 1 summarizes the marine-derived natural compounds reported to exhibit AChE and BuChE inhibitory activity.

### 2.6. Mechanistic Insights into the Multi-Target Potential of Marine-Sourced Bioactive Compounds

Natural substances derived from marine sources exhibit neuroprotective capabilities, as evidenced by various mechanisms of action, not limited to merely inhibiting acetylcholinesterase activity. The primary mechanism through which most substances affect AChE activity is by directly binding to the active site of the enzyme.

In addition, numerous marine metabolites interact with areas of the AChE active site that are not affected by classical medications such as rivastigmine, galanthamine, donepezil, or tacrine. This distinctive binding may facilitate the creation of more effective inhibitors. For example, sargaquinoic acid derived from brown algae *S. sagamianum* exhibits strong inhibitory effects against BuChE, with an IC_50_ value of 26 nM, which is considerably more potent compared to its effects on AChE [53]. A research conducted on *P. chrysogenum* MZ945518 underscores its promise as a marine source of AChEIs, demonstrating a 63% inhibition rate and an IC_50_ value of 60.87 µg/mL for the fungal extract [35]. Molecular docking studies indicated significant interactions between essential metabolites, including 2,3-dihydroxypropyl acetate and palmitic acid, and the active site of AChE. Furthermore, GC/MS analysis revealed the presence of 20 bioactive compounds, many of which exhibit antimicrobial, antioxidant, and AChE inhibitory activities. ADMET profiling validated the favorable pharmacokinetic properties of these compounds, thereby enhancing their potential for drug development.

Certain active compounds derived from marine organisms exhibit supplementary mechanisms of action, distinguishing them as potential agents with multi-targeted effects that influence various pathogenetic mechanisms involved in neurodegenerative processes. Additional mechanisms of neuroprotective action that complement the inhibition of cholinesterase activity are described below.

Amyloid-β aggregation. Certain compounds derived from marine sources function as dual inhibitors, simultaneously targeting both AChE and the aggregation of Aβ, which proves advantageous in the treatment of AD. This dual mechanism may have the potential to decelerate the progression of the disease more effectively. A research focuses on the docking analysis of bioactive compounds exhibiting anticholinesterase properties, evaluating their potential impact on Aβ aggregation [72]. Several compounds and their derivatives that influence both targets—such as sesquiterpene acetate, pyrrole derivatives, plastoquinones, and farnesylacetones—are discussed.Modulation of neuronal survival pathways. A recent investigation into peptides sourced from *Litopenaeus vannamei* shrimp has identified two hexapeptides, QMDDQ and KMDDQ, as effective modulators of the cholinergic system [70]. In PC12 cells subjected to scopolamine-induced neurotoxicity, both peptides significantly inhibited AChE activity and enhanced ACh levels in a dose-dependent manner, with QMDDQ demonstrating superior efficacy. At a concentration of 0.5 mg/mL, QMDDQ significantly reduced AChE activity and increased ACh to 4.98 ± 0.51 μg/mg protein in hippocampal tissue in vivo. This effect was associated with its structural characteristics: two-dimensional correlation–nuclear overhauser effect spectroscopy NMR analysis indicated that QMDDQ adopts an extended spatial conformation with the N-terminal glutamine residue exposed, thereby enhancing its interaction with the AChE active site. Electrostatic considerations further imply that the lack of a positively charged lysine at the N-terminus (which is present in KMDDQ) provides greater conformational stability and bioactivity. Mechanistically, QMDDQ not only directly inhibited AChE but also activated the PKA/CREB/BDNF and protein kinase B (AKT) signaling pathways, leading to a reduction in pro-apoptotic proteins (Bcl-2-associated X protein, Caspase-3) and an increase in anti-apoptotic protein B-cell lymphoma-extra large (BCL-XL) levels. In vivo, the intraperitoneal administration of QMDDQ (30 mg/kg) to scopolamine-treated C57BL/6 mice resulted in improved spatial learning and memory in the Morris water maze, thereby confirming its neuroprotective potential through both enzymatic inhibition and modulation of neuronal survival pathways.Antioxidant properties. Specific marine compounds, particularly those obtained from the seaweed *E. bicyclis*, demonstrate both cholinesterase inhibitory effects and antioxidant properties. The ethyl acetate fraction of *E. bicyclis* exhibited potent inhibitory effects on AChE and BuChE, in addition to notable antioxidant capabilities, which may assist in alleviating oxidative stress linked to neurodegenerative disorders [56]. Fungi sourced from marine environments, including Aspergillus unguis, generate secondary metabolites that demonstrate considerable inhibitory activity against AChE. Among these metabolites are diverse bioactive compounds, such as benzazepin-2-one and derivatives of cinnamic acid, which possess further antioxidant characteristics that enhance their anticholinesterase effects [46].Proteinaceous Venoms. Venoms derived from marine echinoderms, including sea urchins, possess proteinaceous compounds that impede the activity of both AChE and BuChE. Additionally, these venoms include alkaloids, terpenes, and steroids, which play a role in enhancing their enzyme inhibitory properties [73,74].

These mechanisms underscore the varied and powerful characteristics of natural substances derived from marine sources in their ability to inhibit cholinesterase activity, presenting promising opportunities for the creation of novel therapeutic agents aimed at treating neurodegenerative diseases (Table 2).

### 2.7. Structural Insights and Drug Development

Marine natural products frequently exhibit distinctive chemical architectures, including isonitrile derivatives and halogenated sesquiterpenes, which may serve as innovative scaffolds for the creation of cholinesterase inhibitors [75]. Such novel structures have the potential to facilitate the identification of more potent and selective inhibitors.

The variety in the structure of anticholinesterase compounds derived from marine sources has yielded significant insights for the formulation of novel pharmaceuticals. For instance, the marine pharmacophore derived from brominated dipeptides has been utilized to create effective and selective inhibitors of cholinesterase. These compounds demonstrated substantial inhibitory efficacy against both electric eel and human recombinant AChE, exhibiting IC_50_ values ranging from 20 to 70 µM, and were especially potent against horse serum BuChE, with IC_50_ values between 0.8 and 16 µM. Furthermore, the virtual screening of databases containing marine natural products has revealed potential inhibitors of AChE. For example, the molecule CMNPD8714, sourced from the Comprehensive Marine Natural Product Database, was observed to engage with critical residues within the active site of AChE, positioning it as a promising candidate for subsequent development [76].

## 3. Discussion

The data currently available suggest that compounds derived from marine sources offer a distinctive opportunity to broaden the existing range of anticholinesterase agents. In contrast to synthetic medications, numerous metabolites from marine organisms not only inhibit enzymes but also provide additional benefits, including antioxidant properties, anti-inflammatory actions, and modulation of amyloid pathology (Figure 2).

For instance, fungal meroterpenoids such as territrem B and arisugacins exhibit highly effective AChE-inhibition; while peptides sourced from shrimp, like QMDDQ, display a dual mechanism of action that combines enzymatic inhibition with signaling for neuronal survival. Furthermore, marine fish oils that are rich in polyunsaturated fatty acids exemplify how dietary elements can produce synergistic cholinergic and neuroprotective effects.

Nevertheless, several obstacles persist. The majority of research is limited to in vitro experiments or molecular docking studies, with a scarcity of in vivo or clinical confirmations. This deficiency raises concerns about bioavailability, the ability to cross the blood–brain barrier, metabolic stability, and safety profiles. The synergistic effects observed in crude extracts also imply that isolating single compounds may neglect critical interactions, highlighting the necessity for comprehensive pharmacological strategies. Additionally, the discrepancies in reported inhibitory values across various assays emphasize the requirement for standardized methods when assessing marine-derived inhibitors.

Collectively, these insights indicate that marine organisms represent a largely underutilized source of multifunctional neuroprotective agents. Progress in biotechnology, encompassing genome mining, synthetic biology, and nanotechnology, could expedite the identification, enhancement, and delivery of these compounds. Cooperation among natural product chemistry, pharmacology, and clinical neuroscience will be crucial for transitioning from laboratory research to clinical application.

## 4. Materials and Methods

The main aim of this review is to examine and summarize the literature published since 2010 regarding marine-derived natural compounds that demonstrate acetylcholinesterase inhibitory activity, with a particular emphasis on substances sourced from algae, fungi, bacteria, and various marine invertebrates and vertebrates.

This study aspires to clarify the pharmacological mechanisms and therapeutic potential of these compounds in relation to cholinergic deficits linked to neurodegeneration. By evaluating existing research findings, this review aims to create a thorough framework for future pharmacological investigations and drug development that leverages marine resources.

A detailed literature search was conducted across multiple scientific databases, including PubMed, Web of Science, Science Direct, Scopus, and Google Scholar. The search utilized combinations of the following keywords: ACh, AChE, neurodegeneration, dementia, neuroprotection, cholinergic transmission, neuroinflammation, oxidative stress, amyloid beta, tau, marine, algae, fungi, bacteria, invertebrates, and vertebrates. Furthermore, additional relevant studies were identified through a manual examination of the reference lists from significant review articles.

The inclusion criteria focused on the following:(1)Mechanistic studies that investigate various marine sources of substances with AChE-inhibitory activity—specifically their effects on the neurodegeneration process;(2)Research and meta-analyses that assess the efficacy of marine-derived substances.

## 5. Conclusions

Marine-derived compounds exhibiting acetylcholinesterase inhibitory activity encompass a wide array of chemical structures that hold significant promise for the management of neurodegenerative diseases. Their ability to target multiple pathways suggests that these metabolites may influence cholinergic functions alongside other mechanisms that are crucial for neuronal survival and plasticity. The structural variety of these metabolites—spanning phlorotannins, terpenoids, peptides, and alkaloids—demonstrates the biochemical flexibility of marine organisms in adapting to intricate ecological settings, thereby highlighting their potential for multi-target pharmacological applications.

Nevertheless, despite the growing evidence supporting their positive impacts on cholinergic signaling, oxidative stress, inflammation, and apoptotic processes, our current grasp of structure–activity relationships is still incomplete. Additionally, the majority of existing research is derived from in vitro or preliminary in vivo investigations, with systematic assessments of pharmacokinetics, bioavailability, and long-term safety remaining scarce.

Consequently, future studies should prioritize elucidating structure–activity relationships, employing standardized experimental methodologies, and progressing the most promising compounds into translational and clinical trials. Such initiatives will be crucial in identifying which marine-derived substances possess true therapeutic potential and in facilitating their further development.

## Figures and Tables

**Figure 1 marinedrugs-23-00439-f001:**
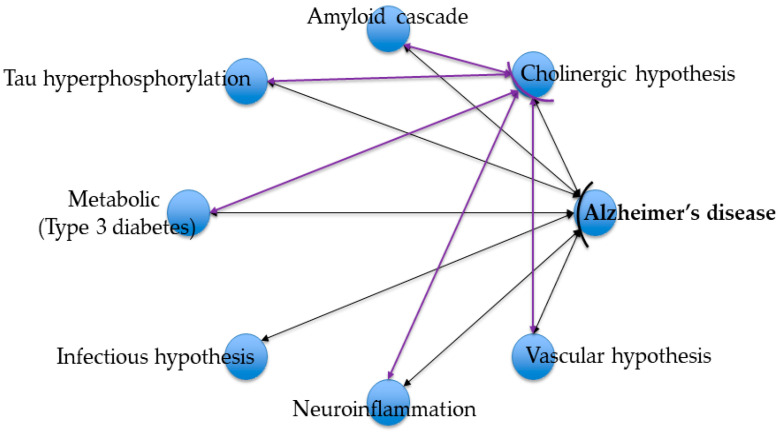
Pathogenetic hypotheses of Alzheimer’s disease.

**Figure 2 marinedrugs-23-00439-f002:**
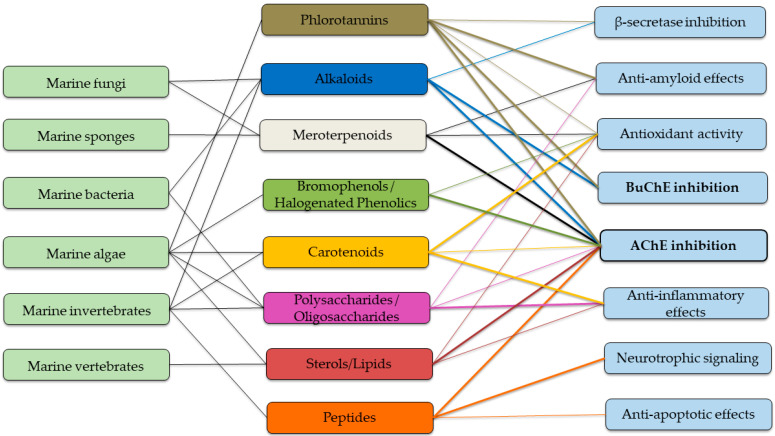
Summary scheme with biological sources, structural types, and mechanisms of action of the marine-derived compounds with AChE-inhibitory activity.

**Table 1 marinedrugs-23-00439-t001:** Marine-derived natural compounds with AChE and BuChE inhibitory activity.

Compound/Extract	IC_50_ (AChE)	IC_50_ (BuChE)	Ref.
6,6′-Bieckol (*Grateloupia elliptica*)	44.5 µM	27.4 µM	[54]
Amphichoterpenoid D (*Amphichorda feline*)	12.5 µM	n/a	[44]
Amphichoterpenoid E (*Amphichorda feline*)	11.6 µM	n/a	
Arisugacin A (*Penicillium* sp.)	1.0 nM	>21,000 nM	[38,39]
Arisugacin B (*Penicillium* sp.)	25.8 nM	>516,000 nM	
Arisugacin C (*Penicillium* sp.)	2.5 µM (2500 nM)	30,000 nM	[39]
Arisugacin D (*Penicillium* sp.)	3.5 µM (3500 nM)	30,000 nM	
Arisugacin O (*Penicillium* sp.)	191 nM	n/a	
*Aspergillus versicolor* metabolite	13.6 µM	n/a	[45]
Aspernigrin A (*Aspergillus niger*)	Active (no value)	n/a	[49]
Aurasperone A *(Aspergillus niger)*	Active (no value)	n/a	
Barettin (*Geodia barretti*)	Active (non-competitive)	n/a	[50]
Cynerine A (*Gracilaria manilaensis*)	Active	Active	[64]
Dayaolingzhiol D (*Ganoderma lucidum*)	8.52 µM	n/a	[47]
Dayaolingzhiol E (*Ganoderma lucidum*)	7.37 µM	n/a	
Dehydroaustin (*Aspergillus* spp.)	0.40 µM	n/a	[40]
Dehydroaustinol (*Aspergillus* spp.)	3.0 µM	n/a	
Dieckol (*Ecklonia stolonifera*)	17.11 µM	Active	[55]
Diphlorethohydroxycarmalol (brown seaweeds)	Predicted active	Predicted active	[65]
Discorhabdin G (*Latrunculia*)	Active (no value)	Selective for AChE	[52]
*Ecklonia maxima* phlorotannins	62.6–150.8 µg/mL	n/a	[58]
Phlorofucofuroeckol-A (*Ecklonia stolonifera*)	4.89 µM	Active	[55]
Eckol (*Ecklonia maxima*; *Ecklonia stolonifera*)	20.56 µM	Active	
*Eisenia bicyclis* extract	Active	Active	[56]
Fish oil (*Haemulon plumieri*)	4.81 µg/mL	n/a	[71]
Fish oil (*Lutjanus synagris*)	2.84 µg/mL	n/a	
Fish oil (*Scomberomorus cavalla*)	2.60 µg/mL	n/a	
Fonsecinone A (*A. niger*, AgNPs)	0.089 µM	n/a	[49]
Graveolinine (*Gracilaria manilaensis*)	Active	Active	[64]
Iso-agelasine C (*Agelas nakamurai*)	30.68 µg/mL	n/a	[51]
Isoaustinol	2.50 µM	n/a	[40]
*Ochtodes secundiramea* extract	400 µg/mL	n/a	[57]
*Penicillium chrysogenum* extract	60.87 µg/mL	n/a	[35]
Peptide KMDDQ (shrimp)	Active (dose-dependent)	n/a	[70]
Peptide QMDDQ (shrimp)	Active (dose-dependent)	n/a	
Phytol (*Gelidiella acerosa*)	2.704 µg/mL	5.798 µg/mL	[62]
Sargaquinoic acid	23.2 µM	26 nM	[54]
Territrem B (*Aspergillus terreus*)	0.071 µM (71 nM)	n/a	[47]

Notes: IC_50_ values are presented as reported in the original studies; “Active” indicates inhibitory activity was observed but no IC_50_ value was provided; “n/a” indicates no data available; “Predicted active” refers to in silico docking or simulation results without experimental confirmation.

**Table 2 marinedrugs-23-00439-t002:** Marine compounds exhibiting neuroprotective properties—groups, representatives, sources, and mechanisms of action.

Compound Group	Representative Compounds	Source Organisms	Mechanism of Action	Ref.
Meroterpenoids	Territrem B, Arisugacin A–Q, Terreulactone A, Dehydroaustinol, Amphichoterpenoid D/E	Marine and endophytic fungi (*Aspergillus*, *Penicillium*, *Amphichorda*)	Potent AChE inhibition (nM–µM), selective for AChE; antioxidant; anti-amyloid	[36,37,38,39,40,41,42,43,44]
Alkaloids	Fascaplysin, Iso-agelasine C, Androtoxin B, Petrosamine, Pulmonarins, Marinoquinoline A	*Marine sponges*, *corals*, bacteria, *jellyfish*	Reversible/noncompetitive AChE inhibition; dual AChE/BuChE inhibition; BACE-1 inhibition; antioxidant	[50,51,67,68,69]
Phlorotannins	Eckol, Dieckol, Phlorofucofuroeckol, Eckstolonol	*Brown algae* (*Ecklonia*, *Eisenia*, *Sargassum*)	AChE and BuChE inhibition (noncompetitive); antioxidant; anti-amyloid; BACE-1 inhibition	[53,54,55,56,57,62,63,64,65]
Peptides	QMDDQ, KMDDQ	Shrimp (*Litopenaeus vannamei*)	Direct AChE inhibition; activation of PKA/CREB/BDNF and AKT pathways; anti-apoptotic and neurogenic	[70]
Polysaccharides/Oligosaccharides	Fucoidans, Chitosan oligosaccharides (COS, DEAE-COS), Sodium oligomannate	Marine algae, crustaceans	Mild AChE inhibition; anti-amyloid and anti-apoptotic; modulation of gut microbiota; anti-inflammatory	[63,66,70,72]
Sterols/Lipids	Fucosterol, Cholesterol derivatives, EPA, DHA	Marine algae and fish (*Scomberomorus cavalla*, *Lutjanus synagris*)	Noncompetitive AChE inhibition; membrane stabilization; antioxidant; anti-inflammatory; anti-tau	[56,66,68]
Carotenoids	Fucoxanthin, Astaxanthin	Brown and red algae, marine invertebrates	Mild AChE inhibition; antioxidant; anti-inflammatory	[66]
Phenolics/Halogenated metabolites	Bromophenols, Halogenated monoterpenes (*Ochtodes secundiramea*, *Rhodomela* spp.)	Red algae	AChE inhibition; antioxidant	[57,60]
Proteinaceous/Venom compounds	Neurosteroidal alkaloids, proteinaceous venoms (*Cassiopea andromeda, Echinometra mathaei*)	Marine invertebrates (jellyfish, sea urchins)	AChE and BuChE inhibition; modulation of Ca^2+^ channels and synaptic function; antioxidant and antiapoptotic	[69,71,72,73,74]

## Data Availability

No new data were created or analyzed in this study. Data sharing is not applicable to this article.

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
