# Peer review of "Marine-Derived Natural Substances with Anticholinesterase Activity"

_marinedrugs, 2025, doi:10.3390/md23110439_

Round 1

Reviewer 1 Report (Previous Reviewer 1)

Comments and Suggestions for Authors

The manuscript “Marine-Derived Natural Substances with Anticholinesterase Activity”, is an interesting and exhaustive review. I consider that the compiled information will benefit the Marine drug researchers.

However, it is suggested that the authors should include more comprehensive information, such as the summarize of the information about the biological sources, structural types, and action mechanism of the compounds in the review. The discussion and conclusion section in the manuscript needs improve to provide a more insightful analysis.

I consider this paper suitable for publication in Marine Drugs upon the following revisions:

  1. In the introduction section, background information of the mechanisms of AChE and BuChE should be presented in the form of figure, which will be more readable.
  2. The chemical structure of the compounds mentioned in the text should be presented, and the common point of chemical structures of AChE inhibitors should be summarized.
  3. All the active data should indicate the positive control and its IC50, so as to facilitate readers' understanding of the strength of the compound's activity.
  4. The content of Figure 2 and Table 2 is repetitive. Please integrate them in a better way.

Author Response

Reviewer 2 Report (Previous Reviewer 2)

Comments and Suggestions for Authors

I appreciate the essential changes made by the authors in the content of the manuscript.

I understand the authors' point of view regarding the content of the manuscript, but I believe that the inclusion of figures with the structures of the mentioned compounds would have made the article more interesting and more accessible to pharmacists and to other specialists for whom the chemical structure is responsible for the properties of substances, including pharmacological ones but also other physical and chemical properties that influence/determine the evolution of a substance in living organisms.

Comments on the Quality of English Language

Author Response

Reviewer 3 Report (Previous Reviewer 3)

Comments and Suggestions for Authors

The authors have carefully addressed all comments and made the necessary revisions.

Author Response

This manuscript is a resubmission of an earlier submission. The following is a list of the peer review reports and author responses from that submission.

Round 1

Reviewer 1 Report

Comments and Suggestions for Authors

Major comments

The manuscript “Marine-Derived Natural Substances with Anticholinesterase Activity”, is an interesting and exhaustive review. I consider that the compiled information will benefit the Marine drug researchers.

However, it is suggested that the authors should include more comprehensive information, such as the summarize of the information about the biological sources, structural types, and action mechanism of the compounds in the review. The discussion and conclusion section in the manuscript needs improve to provide a more insightful analysis.

I consider this paper suitable for publication in Marine Drugs upon the following revisions:

  1. The authors don't mention anything about how they did the research in this review. There are guidelines, such as the Preferred Reporting Items for Systematic Reviews and Meta-Analyses (PRISMA) guidelines, that allow a comprehensive and systematized compilation of scientific literature on a specific topic. Authors must describe the databases used, the keywords, and the criteria for article selection and exclusion. I suggest that authors use the PRISMA guidelines and present the flow diagram of literature search based on the guidelines.
  2. In the introduction section, background information such as the pathogenesis of AD and the mechanism of AChE should be presented in the form of figure or text, so as to prepare readers for a thorough understanding of the content in section 2.6.
  3. The chemical structure of the compounds mentioned in the text should be presented, and the common point of chemical structures of AChE inhibitors should be summarized.
  4. The mixed components whose chemical structures have not been determined (as mentioned in Section 2.1 and Table 2) should not be included in this review.
  5. Table 1 can be replaced by the text and has no need to exist in the manuscript.
  6. The activity values of all active compounds and the positive control should be indicated along with the positive control (for fungal and bacterial sources as well as the compounds listed in Table 3 should be regarded as the key areas).
  7. The examples in Section 2.7 are too few and cannot effectively illustrate the common point of bioactive compounds’ chemical structure.
  8. The discussion section should summarize the information about the biological sources, structural types, and mechanism of action of the compounds in the review through appropriate graphical analysis, making the review more informative and useful.
  9. The Latin name of the species should be in italics. The full name of the species should be used for the first occurrence, and the abbreviation of the species name should be used thereafter.
  10. When a noun first appears, its full name and abbreviation should be displayed simultaneously. From then on, only the abbreviation of the noun should be used.

Detail comments

  1. Check whether the full names and abbreviations of the nouns are used correctly (such as: Alzheimer's disease, acetylcholinesterase, etc.) (Lines 12, 86, 358, 465 …)
  2. Check whether the Latin names are used correctly (Lines 99, 111, 148, 153, 158, 160, 205, 214, 248, 251, 257, 265, 278, 289, 291, 295, 297, 303, 312, 450, 453…).
  3. Line 125:“arisugacin B, arisugacin C, and arisugacin D” should be revised to “arisugacins B–D”.
  4. “Arisugacin” in line 134 should be revised to “Arisugacins”.
  5. Line 246: Extra right parenthesis.
  6. “Sargaquinoic Acid” in line 250 should be revised to “Sargaquinoic acid”.
  7. In line 260, the full name is not displayed for "CH".
  8. Lines 290 and 292: There is a lack of space after the approximately equal sign.
  9. Line 372: "Furthermore, diterpenoids such as gracilin A, which were isolated from marine sponges," should be classified under Section 2.3.

Reviewer 2 Report

Comments and Suggestions for Authors

The scientific names (in Latin) of biological species must be written in italics. Pay attention to the consistency in observing the rule. Cases are indicated by coloring in the manuscript (the attached document).

The chemical name “(Z)-18-octadec-9-enolide, 1,2-Benzenedicarboxylic acid” (Table 1) must be checked.

Lines 124-125: merosesquiterpenes or meroterpenes?

In my opinion, the names of chemical substances in the text must have the corresponding structural formulas in the figures. Otherwise, for many researchers, the content of the manuscript is irrelevant. Cases are indicated by coloring in the manuscript.

Lines 208-212: the fragment requires a bibliographic reference.

Acronyms must be explained in the text, at the first appearance, without exception (GMM, GMMG, etc.). In addition, the list of abbreviations is no longer necessary, especially since it is not complete, and most acronyms are explained when they appear in the text.

Comments on the Quality of English Language

-

Reviewer 3 Report

Comments and Suggestions for Authors

The manuscript marinedrugs-3947341– “Marine-Derived Natural Substances with Anticholinesterase Activity" submitted to journal Marine Drugs brings review on marine bacterial, fungi’s, sponges’ algae’s and other organisms’ metabolites that can be potentially used as inhibitors of cholinesterases.

General comments

  1. The manuscript presents an interesting topic, and based on the title and abstract, I initially expected a well-structured and comprehensive review of the subject. However, the submitted paper does not meet the formal and conceptual standards of a review article. Instead, it reads more like a short research report and therefore requires substantial revision before it can be considered for publication.
  2. Structure and scope

The review is too brief and general, lacking the depth and critical synthesis expected from a review paper. The introduction and sub-sections are overly concise. The authors should expand the introduction substantially, especially regarding Alzheimer’s disease (AD) pathology. It should clearly outline the main biochemical and cellular mechanisms involved, emphasizing that acetylcholinesterase (AChE) represents only one of several potential therapeutic targets. Other enzymes, receptors, and signaling pathways known to play key roles in AD pathophysiology should also be discussed.

  1. References and literature

The manuscript cites an insufficient number of references throughout. The Introduction contains only two references, one of which, cited as “recent,” dates back to 1996. This is inappropriate for a review article. A comprehensive literature update is essential, particularly in the intro sections of every paragraph and descriptive parts of each sub-chapter. The discussion section also lacks proper citation support, even though a review typically does not include a standard discussion; references should still be used to substantiate statements and interpretations throughout.

  1. Formatting and consistency

Latin names of species are not italicized as required by scientific convention. This needs to be corrected consistently throughout the manuscript. The titles and subtitles should be revised to be more specific and informative rather than repetitive variations of the same phrasing.

  1. Tables and content

Table 2 should be cited earlier in the text. Moreover, it is necessary to specify the biological source or organism of origin for each listed compound.

  1. Mechanistic sections

The section describing mechanisms of action is particularly underdeveloped and poorly structured. It is unclear and fragmented across multiple headings. The subsection on “binding sites” is limited to a short paragraph, while the following section on “other mechanisms” combines several unrelated topics without logical flow. The authors should reorganize this part into clearly delineated and conceptually coherent sections, providing detailed explanations supported by appropriate references.

  1. Toxicity considerations

A dedicated section on the toxicity of marine-derived compounds should be included, as there is substantial literature on this topic. Discussing known toxicological aspects would provide valuable context and balance to the review.

  1. Conclusion(s)

The conclusion largely repeats information from previous sections without offering new insights. It should be rewritten to summarize the main points and highlight knowledge gaps, research trends, and future perspectives.

Comments on the Quality of English Language

 The English could be improved.